# Strength and Microstructure Assessment of Partially Replaced Ordinary Portland Cement and Calcium Sulfoaluminate Cement with Pozzolans and Spent Coffee Grounds

**DOI:** 10.3390/ma16145006

**Published:** 2023-07-14

**Authors:** Soorya Pushpan, Javier Ziga-Carbarín, Loth I. Rodríguez-Barboza, K. C. Sanal, Jorge L. Acevedo-Dávila, Magdalena Balonis, Lauren Y. Gómez-Zamorano

**Affiliations:** 1Programa Doctoral en Ingeniería de Materiales, Facultad de Ingeniería Mecánica y Eléctrica, Universidad Autónoma de Nuevo León, Ave. Universidad s/n, Ciudad Universitaria, San Nicolás de los Garza 66455, Nuevo León, Mexico; sooryapushpan@gmail.com (S.P.); javierziga@gmail.com (J.Z.-C.); ivonnerodriguez.lirb@gmail.com (L.I.R.-B.); 2Programa Doctoral en Ingeniería de Materiales, Facultad Ciencias Químicas, Universidad Autónoma de Nuevo León, Ave. Universidad s/n, Ciudad Universitaria, San Nicolás de los Garza 66455, Nuevo León, Mexico; 3Centro de Investigación en Geociencias Aplicadas, Universidad Autónoma de Coahuila, Nueva Rosita 26830, Coahuila de Zaragoza, Mexico; jacevedo7009@gmail.com; 4Department of Materials Science and Engineering, University of California Los Angeles (UCLA), 410 Westwood Plaza, 2121K Engineering V, Los Angeles, CA 90095, USA; mbalonis@ucla.edu

**Keywords:** ordinary Portland cement, pozzolans, spent coffee grounds, calcium sulfoaluminate cement, composite cement, fly ash

## Abstract

Supplementary cementitious materials are considered a viable and affordable way to reduce CO_2_ emissions from the cement industry’s perspective since they can partially or nearly entirely replace ordinary Portland cement (OPC). This study compared the impact of adding spent coffee grounds (SCGs), fly ash (FA), and volcanic ash (VA) to two types of cement: OPC and calcium sulfoaluminate cement (CSA). Cement samples were characterized using compressive strength measurements (up to 210 days of curing), scanning electron microscopy with energy dispersive X-ray spectroscopy (SEM-EDS), X-ray diffraction (XRD), attenuated total reflection infrared spectroscopy, and hydration temperature measurements. In all the studied systems, the presence of SCGs reduced compressive strength and delayed the hydration process. CSA composite cement containing 3.5% SCGs, 30% FA, and 30% VA showed compressive strength values of 20.4 MPa and 20.3 MPa, respectively, meeting the minimum requirement for non-structural applications. Additionally, the results indicate a formation of cementitious gel, calcium silicate hydrate (C-S-H) in the OPC-based composite cements, and calcium alumino-silicate hydrate (C-A-S-H) as well as ettringite in the CSA-based composite cements.

## 1. Introduction

Cement is the most widely used man-made building material in the world, and as the world population grows, demand also surges significantly [1]. Due to global CO_2_ emissions associated with cement production (currently 8–9%) and hence concerns related to climate change, global warming, and ocean acidification, cement manufacturers face the pressure to propose viable and environmentally friendly solutions towards the reduction of carbon footprints [2,3,4,5]. One possible approach extensively explored to reduce cement content is replacing part of the cement with supplementary cementitious material (SCM) such as slag, fly ash (FA), volcanic ash (VA), kaolin, rice husk ash (RHA) as well as other SCMs. Most SCMs are industrial by-products or naturally occurring materials. The incorporation of supplementary cementitious materials [6,7,8,9] not only reduces the environmental impact and overall cost of cement production but can also boost material properties, e.g., enhance workability, durability, and strength in concrete. Through pozzolanic reaction, silica-rich SCMs can increase the formation of certain cement hydrates such as the C-S-H phase and can hence reduce porosity and improve the durability of cement pastes. Alumina-rich SCMs are reported to promote the uptake of Al by C-S-H, resulting in C-A-S-H formation [10]. Alumina-rich SCMs could also result in the formation of hydrotalcite and AFm phases when a low sulfate-to-alumina ratio is attained and Mg is present [11,12].

Calcium sulfoaluminate cement (CSA) is produced at 1250 °C, which is about 200 °C less than a typical OPC, and with 25–30% less in net CO_2_ release [13,14]. Its benefits are not only related to energy savings and low carbon emissions [15,16] but also to quick setting and early hydration that increase strength gain in the early curing periods. These so-called rapid-set cements can be conveniently utilized in construction projects where time is of the essence, such as repairs of busy highways or airports, which can be closed only for a limited period [17]. Ye’elimite (C_4_A_3_Ŝ), the primary phase detected in a CSA clinker, is present in amounts ranging from 30 to 70 percent by weight, and its hydration yields the rapid formation of ettringite (AFt), which is the main hydration product [18]. In addition, because of ye’elimite’s hydration, aluminum hydroxide and monosulfate (sulfate-AFm) are formed, while aluminum hydroxide is usually found in amorphous structures [19]. In addition to aluminum hydroxide, non-expansive large ettringite crystals, which exhibit a prismatic habit, contribute to high mechanical strength. Essential characteristics of the hydration process are linked to a high reaction rate and stability, a considerable consumption of free water, and a reduction of capillary porosity [18]. Other hydration products like C-S-H, monocarbaluminate (carbonate-AFm, due to presence of atmospheric CO_2_), strätlingite, and gibbsite can also precipitate because of the existence of minor phases such as calcium aluminate ferrate, gehlenite, excess anhydrite, and belite [20].

Volcanic ash is a naturally available material abundant in alumina and silica contents. Milling and transportation of this material require significantly less embodied energy, lower than that needed for cement production, which could reduce carbon footprints [21,22] when using this material to partially replace the cement. Introducing VA replacements into concrete has been used with the aim to increase mechanical strength and minimize concrete production costs [23,24]. Also, the alumina introduced via VA was found to substitute into the C-S-H, which could enhance the resistance to sulfate attacks [25,26].

Fly ash is a by-product of the coal-fired electrical industry. It is known to improve compressive strength in cement pastes by generating more C-S-H through the pozzolanic reaction with portlandite [27]. Owing to the high alkalinity of the cement slurry, FA can dissolve and interact with the portlandite to produce C-S-H [28]. In addition, the inclusion of FA enhances the workability of cement or concrete. Moreover, there is a filler effect due to the extra nucleation sites on the surfaces provided by this SCM. Several reports indicated that by the addition of FA, segregation, bleeding, heat evolution, and permeability are reduced, inhibiting the alkali-aggregate reaction and enhancing sulfate resistance [29,30].

Coffee is regarded as the most consumed refreshment drink globally, with a total green coffee bean production of 9.95 million tons [31]. On the other hand, spent coffee grounds (SCGs) are considered a significant municipal waste from households and cafes and are meant to be landfilled. Teck-Ang Kua reported that developing a derivative construction material which can incorporate spent coffee grounds will aid the diversion of this insoluble solid waste from landfills [32]. Fernanda Andreola [33] reported that 15 wt% of SCGs mixed with clay can be used to manufacture high-quality lightweight clay aggregates for drainage functions. Leopold Lee Poh Chung studied SCGs and tea waste to prepare alkali-activated bricks, which could be effectively utilized as an additive to manufacture environmentally friendly raw clay bricks. Samples that contained 2.5 wt% SCGs and 10 wt% tea waste yielded a compressive strength of 8.6 MPa, therefore meeting the criteria for construction operations. However, it has been noted that SCGs exceeding 5 wt% can lead to the formation of fungus, affecting the durability of the structure due to its slow strength development [34]. Geopolymer prepared using SCGs and blast furnace slag at 50 °C can be used as a pavement-filling material [35]. According to the literature, very few studies have been carried out focusing on mechanical strength performance and microstructure analyses based on composite cement prepared using SCGs and other pozzolans.

On the other hand, from 1996 to 2004, J I Escalante-Garcia and Sharp conducted various studies on blended cements, OPC, GGBS, FA, and VA [36,37,38,39]. They combined OPC with 60% blast furnace slag, 30% FA, and 22% VA, curing the mixture at temperatures between 10 and 60 °C. Their investigations found that the porosity of the composite cement increased along with the curing temperature [40]. The hydration rate was more remarkable for lower curing temperatures (at 10 °C), and a lower porosity was observed. The C-S-H formed in the composite cement prepared with VA showed higher Si and Al contents and a lower amount of Ca. Al-Fadala et al. [41] studied cement blended with VA in a range from 10 to 40% of replacement levels and found that it helped to reduce heat release during the hydration process as the replacement percentage increased. In the initial days of curing, the compressive strength was reduced by nearly 40% when the OPC was replaced with VA. Still, it gradually increased in the later curing periods, indicating that the reaction in the composite cement was more significant at later ages.

Various studies were carried out on blended cements containing OPC and FA. The partial replacement of OPC by FA (20–30% by weight) improved mechanical properties. Precipitation of the C-S-H gel enhanced the strength of the composite cement mix. A reduction in porosity and a decrease in the average pore size were associated with the fineness of the FA [27,42,43,44,45,46,47,48,49,50].

Experimental studies on the retarding effect of SCGs were carried out at various temperatures by Rahul [46]. The introduction of SCGs increased the setting time of the composite cement slurry. To reduce the time needed for the setting and hardening, the curing temperature must be increased [51]. The synthesis of geopolymer pastes with SCGs, rice husk ash (RHA), and blast furnace slag (S) for pavement applications was reported elsewhere. This research revealed that 70:20:10 was the ideal mix ratio for SCGs: RHA:S, which reached its maximum compressive strength when curing was performed at 50 °C [52]. The strength assessment of a geopolymer mix prepared using SCGs, slag (S), and FA was analyzed by Teck Ang Kua, and the optimum mix recommended contained a weight percentage of 70% of SCGs and 30% of S with a liquid-to-solid ratio of 55%, which was recommended as a pavement filling material for road embankment purposes [32].

As alternatives to OPC, CSA mortars and CSA FA-based alkali-activated mortars were studied in the past to evaluate their hydration behavior and their physico-mechanical characteristics [53,54,55]. Composite cement prepared using CSA and FA showed better mechanical strength due to the rapid formation of ettringite. Moreover, CSA with FA hydration studies were conducted on mortars at different curing ages. These findings support the idea that the partial replacement of calcium sulfoaluminate with FA positively affects the environment and the economy [56]. Martin et al. studied the addition of FA to calcium sulfoaluminate cement, where, with the increase in the FA content, CSA hydration was accelerated mainly due to the filler effect of FA. The addition of 7.5% FA resulted in the highest compressive strength, and the maximum amount of FA that may be used without losing strength was 15% by mass [28]. Additionally, eco-friendly cement containing CSA, blast furnace slag, and silica fume were prepared for structural work, as reported elsewhere [57].

Considering all of the above, it was concluded that there is a lack of information on the properties of composite cements containing VA, CSA, and SCGs in terms of both structural and non-structural applications. Following from that, this work is focused on comparing the mechanical strength and microstructural properties of two types of composite cements based on the OPC and CSA prepared with SCGs, FA, and VA. The aim of this study is an overall reduction in the use of OPC and hence the mitigation of CO_2_ emissions as well as the utilization of waste materials such as SCGs to decrease landfill disposal and hence environmental impacts.

## 2. Materials and Methods

Ordinary Portland cement (OPC), from CEMEX, Monterrey, Nuevo Leon; calcium sulfoaluminate cement (CSA), from Cemento Fraguamax of Cementos Chihuahua; fly ash (FA), provided by CEMEX; volcanic ash (VA), from Vulkano Block Termico, Monterrey, Nuevo Leon; and spent coffee grounds (SCGs), from Mexico (same type of coffee) were used in this research. The FA was sieved through mesh no. 200 (75 microns) to obtain the exact particle size distributions of all raw materials. ASTM 0402 C311 was used to optimize the fineness of the pozzolans [58]. The VA was subjected to ball milling for 90 min and then sieved through the mesh. ASTM 0401C595 standards were used for the preparation of the volcanic ash [41,58,59]. SCGs were collected and allowed to dry at 100 °C for 24 h and then sieved through the mesh. Table 1 displays the XRF chemical composition and density of each raw material used.

### Characterization Techniques

(a)Mechanical properties assessment: Preliminary tests were carried out to optimize parameters such as the setting time, water–cement ratio, and workability of the composite cement pastes. For optimizing the w/c ratio, OPC- and CSA-incorporated composite cements were prepared separately, each with FA, VA, and SCGs at different w/c ratios (0.4, 0.45, and 0.5). It was found that FA and VA blends require comparatively less w/c ratios than the ones made with the SCGs. Also, increasing the w/c ratio for the composite cement samples FA and VA decreases the compressive strength (CS). Using a w/c ratio of 0.4, a shrinkage in the hardened state for the FA- and VA-incorporated composite cements was observed, while at the same w/c ratio in the SCG-supplemented cement, the samples required extra water to make them workable. Raising the w/c ratio to 0.5 decreases the CS results for all composite cements made with FA, VA, and SCGs and shows a delay in setting and hardening for composite cements prepared with SCGs. At a w/c ratio of 0.45, all the composite cement samples were workable, and the VA showed better CS results as compared to the FA and SCGs. Thus, the water-to-cement (w/c) ratio was fixed at a 0.45 ratio. Composite cement cubes of 25 mm were prepared and cured under a calcium hydroxide-saturated solution for 3, 7, 28, 90, and 210 days. Five badges of samples were designed. The first badge of samples was prepared with FA at a 0, 10, 20, and 30% weight replacement level of the cement. The second set contained VA at a 0, 10, 20, and 30% weight replacement level of the cement. The third badge contained a 0, 1.5, 2.5, and 3.5% weight replacement level of the cement made with the SCGs. The fourth was a combination of 30% FA and 3.5% SCGs. The fifth was a combination of 30% VA and 3.5% SCGs. ASTM 401 C305 and ASTM 401 C109 [60,61] standards were used to execute the mixing procedure and the mechanical strength measurements, respectively. The effect of the curing temperature was also evaluated for samples containing SCGs, which were cured at 25 and 80 °C for 24 h. The samples containing SCGs were difficult to set at room temperature, so after several preliminary tests, an 80 °C temperature was chosen to enhance the setting. Compressive strength tests were performed after 3, 7, 28, 90, and 210 days of curing.(b)X-ray diffraction (XRD): Selected samples were crushed and kept in isopropyl alcohol for three days to stop further hydration. These samples were then dried at 40 °C for 48 h, ground, and sieved through 75 microns to obtain powdered samples for the XRD analysis. A Bruker-D8 Advance diffractometer with a Cu-Kα radiation source (λ = 1.54) was used to analyze the mineralogical phases within a range of 2θ between 5 and 70° with a step size of 0.021 and a scan dwell time of 1 s [62].(c)Scanning electron microscopy and energy dispersive electron spectroscopy (SEM-EDS): Samples that showed better mechanical strength were chosen for morphological and structural analyses. These samples were kept in isopropyl alcohol for 48 h to stop the hydration and then dried at 40 °C for three days. Subsequently, these samples were mounted, polished, and gold coated for SEM and EDS analysis. The morphology and elemental composition of each sample was analyzed on SEM with EDS-attached equipment in the back-scattered electron mode (BSE), applying an accelerating voltage of 10 kV.(d)Attenuated total reflectance infrared spectroscopy (ATR-IR): IR spectra were collected for composite cement cured for 90 days, using a BRUKER Alpha II spectrophotometer. The same powder preparation procedure was applied as for the XRD measurements. The spectrum was collected by averaging 32 scans between 2000–400 cm^−1^ with a 4 cm^−1^ spectral resolution.(e)Measurement of the hydration temperature: Each blended cement was mixed thoroughly with water and poured into a thermal disposable glass to study the hydration temperature. Copper constantan thermocouples were inserted into the cement paste and subsequently placed in a double-walled box, which was tightly sealed. Thermocouples were connected to a potentiometer, which recorded the temperature of the cement paste every second for a period of 24 h. The LabVIEW 2011 program was used to plot the hydration temperature vs. time graph for each of the studied cement pastes [63,64,65].

## 3. Results and Discussion

### 3.1. Compressive Strength

Table 2 shows the mixture designs of the cement samples containing OPC. Figure 1 provides the CS results of the composite cements described in Table 2 after 3, 7, 28, 90, and 210 days of curing. Figure 1 shows that the mechanical strength of the composite cements was improved at later ages. Apart from chemical composition, physical properties such as an increased fineness of the FA and VA, the optimum water-to-cement ratio, mix design, and extended curing period can influence the strength of composite cements. The composite cements prepared with OPC and SCGs required elevated temperatures to set and harden. Jadhav et al. [51] also reported that setting is difficult to achieve in cement samples containing SCGs without the help of temperature, and as a result, CS values dropped as a function of the increasing SCG content. Preliminary results showed that the increase in the SCGs resulted in a negative effect in terms of cement setting. The maximum content of SCGs was therefore set at 3.5% of the replacement level of cement. After 210 days of curing, the CS of the composite cement containing SCGs with and without and the aid of a high curing temperature were measured at 26.1 MPa and 24 MPa, respectively. These values are higher than the other reported results for samples comprising cement and SCGs [32,52,66]. Replacing OPC with SCGs requires more water as compared to the SCM, which could affect the workability of the mix. The w/c ratio was fixed based on preliminary studies, and for a value of 0.45, the samples exhibited better mechanical properties in terms of compressive strength. No shrinkage was observed in the hardened state compared to cement samples prepared with a w/c of 0.4. For lower w/c ratios, cement samples prepared with SCGs showed a lower workability. An increase in the w/c ratio to 0.5 resulted in poor setting and hardening for the SCG-incorporated composite cement samples and decreased mechanical strength. This was also one of the reasons why the optimum replacement percentage of the OPC with SCGs was set at 3.5%. While a 30% replacement level of the OPC with VA showed a reduction in the CS values at earlier ages, i.e., up to 28 days, after 210 days of curing, the CS results were enhanced, with a recorded value of 45 MPa [24,67]. Siddique et al. reported similar CS findings for the OPC blend containing VA. The fineness of the VA also resulted in the improved CS for the VA–OPC cement. The surface area of particles increases as the particles become finer, and a larger surface area enhances the reaction rate. When the material is finer, there is more potential for the pozzolanic reaction of VA, which results in a lower porosity and better CS outcomes [26,68]. Analyzing the CS data already reported for cements composed of OPC and VA and comparing them to the results collected herein, it was found that the CS data obtained in this work were higher, i.e., 42.9 and 45 MPa after 90 and 210 days of curing, respectively [67,68,69,70]. The optimum replacement level of the OPC with FA was also established at 30%, similar to VA. The CS for the composite cement, which incorporated OPC and FA, was 42.2 MPa after 210 days of curing. These results are consistent with the findings reported in previous investigations, where similar compressive strength values were outlined [71,72,73,74]. Workability was improved in the presence of FA due to its spherical morphology. While comparing the already reported CS values, the values obtained in the present study were higher, 40.4 and 42.2 MPa after 90 and 210 days of curing, respectively [71,75]. This was attributed to the fact that, at later ages, the formation of the reaction products is enhanced in both VA and FA blends due to the pozzolanic reaction, which resulted in the formation of secondary C-S-H densifying the composite cement matrix [26,27,30,69].

The CS values decreased when compared to the CS results of the SCG-incorporated composite cement. This is associated with the fact that the SCGs served as a retarder [51,76]. As per the nucleation theory, the retarder binds to the nuclei of hydration products to reduce their rate of reaction and overall development. By occupying the reactive sides, the retarder primarily prevents the hydration of cement.

Table 3 shows the mixture designs for the CSA-based cement composites. The CS results obtained for these cements are shown in Figure 2. It was noted that in the case of the CSA, supplementary cementitious materials (i.e., VA and FA) showed better results at later ages. It was also observed that the hydration reaction could be retarded by the presence of VA and FA since, at early ages, 10% and 20% replacement levels of CSA yielded higher CS values as compared to the 30%. However, after 28 days, the samples made with 30% of VA and FA showed higher CS values than the 10 and 20% replacement levels. Nordine Leklou also reported a similar trend of increase in CS results for higher replacement levels of cement prepared with FA [74]. The CS results obtained in this work are slightly higher than the ones previously reported for similar blends. After 28 days of curing, the SCF30 studied in this work reached 34 MPa, while Martin et al. reported a strength of 29 MPa for a similar composition [28].

Additionally, it was observed that the CSA blend which hydrated at room temperature showed similar strength as the one cured at 80 °C. The hydration reaction in the CSA is more exothermic than the one in the OPC, so it does not require temperature assistance to aid setting. Replacement of the CSA with 30% VA and 30% FA yielded a CS of 53.6 MPa and 43 MPa, respectively, after 210 days of curing. The CSA containing SCGs showed a value of 28.9 MPa after 210 days when hydrated at room temperature, and its strength decreased to 26.4 MPa as a function of the increasing curing temperature. The CSA comprising the VA and SCGs showed a value of 21.8 MPa, and the CSA containing FA and SCGs yielded 21 MPa at room temperature, and their strength decreased to 20.3 MPa and 20.4 MPa, respectively, when the same blend was cured at 80 °C. Without temperature aid, the samples prepared with 2.5% and 3.5% of SCGs had almost identical CS values of 27.8 and 27.9 MPa, respectively. CSA is more water-demanding than OPC; hence, the CSA composites containing SCGs require higher water-to-cement ratios. Nevertheless, for the fixed water-to-cement ratio of 0.45, the CSA blends showed high compressive strength values.

A comparison of Figure 1 and Figure 2 shows that VA performs better with CSA as compared to the OPC. In addition, composite cement containing SCGs performed better in the CSA systems than in the OPC ones. In the case of FA, the blends containing FA produced almost similar compressive strength values in the OPC and the CSA, 42.2 MPa, and 43 MPa, respectively. The OPC and CSA prepared with SCGs presented similar results in terms of strength, where OCC3.5 and SCC3.5 resulted in a value of 26.1 MPa and 28.9 MPa, respectively. The composite cement OCFC yielded 20.2 MPa, while SCFC showed a value of 21 MPa. For the composite cement, the OCVC and SCVC strength was estimated at 20.4 and 21.8 MPa, respectively. SCGs additionally did not require any temperature aid for setting and hardening, indicating another advantage of using CSA over the OPC.

Figure 3 compares the compressive strength (CS) results for samples comprising the OPC or CSA and containing FA and VA at different curing days. The CS results of the CSA composites cured at 210 days showed slightly higher values than those for the OPC, 55.4 MPa, and 53.2 MPa, respectively. The composite cement made of CSA blended with FA and VA presented better CS values than the OPC mixtures. SCF30, SCV30, OCF30, and OCV30 had CS values of 43, 53.6, 42.2, and 45 MPa, respectively. Even if the CS results of the composite cement designed with CSA did not exceed the CS values of the reference sample SC0 cured at 210 days, the results revealed that a comparable CS value was obtained for the neat CSA paste and SCV30 of 55.4 MPa and 53.6 MPa, respectively. The results also demonstrate that a 30% substitution of the CSA with VA did not significantly affect the CS result of SC0. SCV30 exhibited low CS values in the initial curing days at 3, 7, and 28 days of curing, but it reached almost the same result as that of OC0 at 90 and 210 days. The CS results, therefore, indicate that VA-incorporated CSA composite cement could be a better substitute for the OPC in terms of mechanical properties’ development.

### 3.2. X-ray Diffraction

X-ray diffraction patterns of the OPC-based composite cements cured for (a) 28 days and (b) 90 days are shown in Figure 4. The diffraction patterns displayed alite, calcite, ettringite, monosulfate (sulfate-AFm), and portlandite, generally seen in hydrated OPC systems [77,78,79,80]. The main hydration product of the OPC is calcium silicate hydrate (C-S-H), which is usually described on XRD as an amorphous halo in the 2θ range of 24° to 36° [77,78], calcium hydroxide (CH) termed portlandite (P), Aft, and AFm phases. The most encountered AFt phase is ettringite, and in the case of the AFm phase, monosulfate. The hydration reaction of alite and belite in the OPC leads to the formation of C-S-H and CH [81]. Also, a reduction in the alite intensity peak is related to the progression of the hydration reaction and the resulting formation of C-S-H and CH phases [82]. In blended cements containing VA, FA, and SCGs, hydrated phases typically formed in the OPC systems along with the presence of quartz (which remained unreacted) were identified. Garcia Mate also reported unreacted quartz for a similar system of OPC supplemented with FA [56].

Blends OCF30 and OCV30 showed a decrease in the portlandite intensity, suggesting a pozzolanic reaction at later ages, which is in line with the CS results. [83,84,85]. Additionally, unreacted phases were observed in the composite cement containing SCGs, indicating that the presence of SCGs retarded the hydration reaction and hence adversely impacted the mechanical strength (see Section 3.1). The monosulfate peak identified at 2θ of 11.7° was only found in OC0, OCF30, and OCV30. The presence of the monosulfate phase was not observed in the OCC 3.5, OCFC, and OCVC blends, suggesting that the incorporation of SCGs influenced the formation of hydrates at the early hydration stage. The rapid dissolution of SCGs in water builds up the viscosity of water; thus, the migration rates of ions such as Ca^2+^, SO_4_^2−^, and OH^−^ were reduced. In addition, the adsorption of SCGs on cement particles leads to the retardation of the hydration reaction, i.e., formation nucleation, growth, and subsequent precipitation [86].

No other phases were found even after 90 days of curing, which indicates that the strength development in the composite cement was mainly associated with the C-S-H formation. A delay in the initial formation of ettringite in the composite cement was also observed, with an increase of the peak intensity of that phase at later ages. As a result, the CS diminished with an increase in the replacement level.

The diffraction patterns for the CSA-based composite systems after 28 and 90 days of curing are shown in Figure 5. Unhydrated CSA contains ye’elimite (C_4_A_3_Ŝ) as the main constituent. The hydration reactions between C_4_A_3_Ŝ, calcium sulfates (anhydrite), and water initiate rapid setting and the formation of ettringite (AFt), which is the main reaction product. Ettringite contributes to the accelerated strength development in the CSA system by reducing the porosity of the cement matrix since ettringite has a relatively large molar volume [87]. The diffraction pattern disclosed the presence of larnite, calcite, ettringite, ye’elimite, and strätlingite. Ettringite was observed at 2θ angles of 9°, 15.7° and 22.89° 2θ [12,56]. In addition, several unreacted phases were still observed in the hydrated CSA, but their intensity reduced as the hydration progressed. Unreacted calcite was observed due to its limited solubility in water. Al(OH)_3_ was not observed either in the reference or composite cement samples, indicating that if Al(OH)_3_ had formed as published elsewhere, it was amorphous [20]. As expected, the decrease in the intensity of the ye’elimite peak corresponded to an increase in the ettringite phase formation under long-term curing conditions [88].

The peak of quartz was found unaltered after 90 days of curing, indicating that it did not take part in the hydration reaction, which is the same result as in the case of the blended OPC cement [56]. Ye’elimite peaks were observed at 2θ of 23.65°, 40.9°, and 62°. The CSA-VA composite cement showed a lower intensity of this phase, compared to the neat CSA, due to the hydration reaction of VA with the CSA, which resulted in the formation of ettringite. Correlating these findings to the compressive strength results, it was observed that SCV30 showed better compressive strength results, supporting the claim that VA exhibited a higher reactivity than FA.

After 28 and 90 days of curing, strätlingite (S) was identified. The reaction of larnite, aluminium hydroxide, and water led to strätlingite (aluminosilicate AFm) formation, which is known to require longer times to form in the cementitious system [20,56,87,89,90]. Formation of strätlingite helped to generate a denser microstructure at later hydration ages, hence reducing the overall porosity. Strätlingite was formed and observed at 2θ angles 7° and 14.2° in the SCF30 and SCV30 samples, due to the presence of reactive amorphous aluminosilicates supplied by FA and VA [28]. Nevertheless, strätlingite was also observed in all other cement samples, including the reference sample, just with a lower intensity.

### 3.3. Scanning Electron Microscopy and Energy Dispersive Electron Spectroscopy

Figure 6 shows the SEM-SE images at 4000× magnification after 90 days of curing for the neat OPC, OCC 3.5, OCFC, and OCVC. Hydration products such as C-S-H, ettringite, and portlandite were observed in these images. C-S-H, which, due to its poorly crystalline structure, could not be identified in the XRD results, was visible in the SEM images. The ettringite, exhibiting a needle-like structure, and portlandite, which has a laminar structure, were found in these images [91,92,93,94]. AFm phases, which derive from portlandite, also exhibit a layered structure [95,96,97]. Due to these similarities in morphology, it was impossible to distinguish the AFm phase from the portlandite. The portlandite observed in the SEM image of the OC0 sample was more prominent as compared to the composite cements. Portlandite was observed for the composite cement in a smaller proportion, indicating that a portion of the portlandite reacted with the FA and VA via pozzolanic reaction to form C-S-H, as previously reported [98]. Unreacted FA particles can be seen in the SEM image of the OCFC sample, in Figure 6c.

Figure 7 shows the SEM-BEI images and corresponding EDS spectra of the samples cured at 80 °C for (a) OCC 3.5, (b) OCVC, and (c) OCFC. In all the systems, some unreacted particles were found (off-white color), with some rings (gray) around those unhydrated grains displaying specific characteristics of the C-S-H gel. These partially hydrated cement particles and a C-S-H type gel are grey due to their water content [99]. Porous regions are indicated by darker areas and are dispersed throughout the matrix. A homogenous matrix with minimal porosity also resulted in a high mechanical strength. Figure 7a shows the SEM-BEI images and corresponding EDS spectra of the composite cement containing OPC and SCGs. Each cementitious matrix was individually analyzed, considering the different zones of the gel, since these regions are typically poorer or richer in different types of oxides, as reported in Table 4. The results disclosed high CaO and SiO_2_ contents and small quantities of Al_2_O_3_ and Na_2_O for the composite cement samples made based on the OPC. The Na_2_O content was elevated in the samples that incorporated SCGs. The SEM-BEI image and EDS spectra of the OPC and composite cement samples made with 30% FA, 3.5% SCGs, and OPC and with a temperature aid is shown in Figure 7b. This SEM-BEI image showed unreacted raw materials such as FA, SCGs, OPC, compact hydrated cement matrix, and some porous regions. The XRF data presented in Table 1 show that FA is rich in SiO_2_ and has significant amounts of Al_2_O_3_. The oxide percentage calculated from the EDS analysis confirms that the gel formed in the composite cement is C-S-H (with a small quantity of aluminum oxide) with the following oxide ratios: CaO/SiO_2_ = 1.15, Al_2_O_3_/SiO_2_ = 0.26, and Na_2_O/SiO_2_ = 0.07. Studies by Barbara Lothenbach and García-Lodeiro also reported C-S-H gel formation in the composite cement made of the FA and OPC [10,79].

The SEM-BEI image of the sample containing VA, OPC, and SCGs set at a high temperature is shown in Figure 7c. The composite cement made with VA showed a compact cement matrix and porous regions. VA contains high quantities of SiO_2_, similarly to FA, as shown in Table 1. The EDS analysis showed the additional formation of the C-S-H gel (also along with the presence of aluminum oxide) in this composite cement. When comparing the SEM-BEI images in Figure 7a–c, more compact hydration products were observed for the composite cement systems: Figure 7b, OCFC and Figure 7c, OCVC. The FA and VA in each composite cement mix can act as micro fillers, decreasing porosity in these systems. SCGs were found to retard the formation of hydration products; thus, SCGs primarily prevent the formation of highly reactive nano nuclei of cement hydrates which, in consequence, is believed to be the principal reason for the low compressive strength in samples containing SCGs [51]. For the OPC composite cement systems, the EDS results in Table 4 showed that the matrix was composed mainly of Ca, Si, and Al, indicating that the matrix is made up of a cementitious gel rich in calcium and silica, i.e., C-S-H [23].

At higher temperatures, there is a possibility of transforming amorphous C-S-H into crystalline analogs like jennite, tobermorite, and afwillite structures [81,100]. Usually, tobermorite is observed as a layered structure, and jennite is observed in lath structures. The Ca/Si ratio will also be lesser for the crystalline phases of C-S-H (tobermorite, jennite, and afwillite) when compared to the amorphous C-S-H [80,101]. In this case, a higher temperature (80 °C) was used for the samples which incorporated SCGs, because SCGs require a higher temperature to set, harden, and better react with the cement used, as reported by Jadhav [51]. The compressive strength results show a decrease in the mechanical strength in the samples containing SCGs and cured at room temperature. From the SEM images, it can be observed that C-S-H shows distributed typical structures for all the samples, i.e., for OC0, OCC3.5, OCFC, and OCVC. The results suggest that there was no significant change in the C-S-H structure at a higher temperature. From the EDS results, the Ca/Si ratio was like the one found in jennite for the samples OCC3.5, OCFC, and OCVC. These three samples were prepared with SCGs and other pozzolans such as FA and VA. The presence of pozzolans also depresses the Ca/Si ratio [98,102]. The experiments by Bauchy [103] reveal the possibility of transforming a part of amorphous C-S-H to its crystalline analog. Nevertheless, the XRD results did not disclose any crystalline tobermorite or jennite. Ettringite was formed during the initial stage of the cement clinker hydration, and it decomposed when it was exposed to high temperatures.

Ettringite regenerates as concrete returns to room temperature and absorbs water from the surrounding environment. When ettringite is formed within a hardened concrete body, it can cause expansion pressure, resulting in cracking, known as delayed ettringite formation (DEF) [101,104]. Many other factors affect the DEF other than high temperature, such as the addition of chemical admixtures, the influence of pozzolans, the size and fineness of pozzolans used, humidity conditions, and the pH. The fineness and Al_2_O_3_ content of the pozzolans used in this work (FA, VA, and SCGs) could reduce the possibility for DEF expansion [105]. From the EDS results, the SO_3_/Al_2_O_3_ molar ratio obtained for the composite cement was less than 1. This also confirms the lower possibility of DEF in the SCG-incorporated temperature-assisted samples [106,107]. The literature review performed by Yulu Zhang explains a detailed study of this phenomenon for samples treated at higher temperatures [108]. Although there was no cracking observed in the samples cured up to 210 days, studies at longer curing periods must be carried out to confirm this hypothesis.

The SEM-SE images of pure CSA and composite cement containing CSA, SCC3.5, SCFC, and SCVC, cured for 90 days are shown in Figure 8, at 4000× and 11,000× magnifications. Hydration products typically formed in this type of composite cement were described in detail elsewhere [89,109]. The main hydration product of CSA was ettringite, as observed on Figure 8 and the XRD results, and in agreement with the literature [110,111]. Small amounts of C-S-H were observed in the SEM images. The EDS results confirm the formation of C-S-H containing aluminum, suggesting C-A-S-H precipitation [112,113]. Ettringite, which has a needle-like structure, was visible in all the CSA systems analyzed in this work. Like the OPC-based blends, it tends to grow in the pores of the cementitious matrix. The ettringite formation can be observed in the SCFC samples in Figure 8e,f. Nevertheless, ettringite needles were more prominent in the SCC 3.5 sample, as observed in Figure 8d, and the intensity of the ettringite peak was also more intense in the XRD spectra.

The SEM-BEI images and the corresponding EDS spectra of the composite cement samples made with CSA and cured for 90 days are shown in Figure 9. The results in Table 4 show that SCC3.5 displayed higher CaO and Al_2_O_3_ contents due to the initial chemical composition of the raw materials (Table 1), with the following initial ratios: CaO/SiO_2_—3.22, Al_2_O_3_/SiO_2_—2.77, and Na_2_O/SiO_2_—0.01. Higher ratios of CaO/SiO_2_ and Al_2_O_3_/SiO_2_ are disclosed in Table 4, resulting in the formation of the C-A-S-H gel [20].

Figure 9b shows the SEM-EDS images of the composite cement made with CSA, FA, and SCGs. The image shows unreacted spherical particles of the morphology typical for FA, and the irregular shapes belong to the SCGs and CSA. The region near the cenospheres of the FA showed a high amount of CaO and Al_2_O_3_, which was associated with the C-A-S-H gel [79,114,115]. Table 4 shows a high percentage of Al_2_O_3_ and ratios of CaO/SiO_2_ = 3.07 and Al_2_O_3_/SiO_2_ = 2.54, indicating the formation of the C-A-S-H gel [99]. Formation of the C-A-S-H gel was also reported by Martin and Sánchez-Herrero in their studies focusing on the CSA-FA blends [28,114]. The microstructure of the sample containing CSA, VA, and SCGs is shown in Figure 9c. Unreacted VA particles are identified as dark grey with an irregular morphology. The EDS analysis also revealed other regions rich in Ca, Si, and Al, which are associated with the C-A-S-H forming because of VA addition which contains a high amount of Si and Al, like FA [99,116].

Figure 9a shows that SCC3.5 has more porous regions as compared to the other two composite systems. Unreacted SCG particles can be observed in this figure which indicates that SCGs have low activity and hence do not improve strength, as confirmed by the CS measurements. Nevertheless, the composite cement systems (Figure 9b) SCFC and SCVC (Figure 9c) produced more compact hydration products, which helped to fill the micropores of the cement matrix, hence reducing porosity and increasing CS.

Figure 7 and Figure 9 disclose that the CSA composite cements have a denser structure as compared to their OPC equivalents, indicating that more hydration products with fewer porous regions (and corresponding higher CS results) were observed in the CSA systems.

### 3.4. Attenuated Total Reflectance Infrared Spectroscopy

Figure 10a shows the ATR infrared spectra of the hydrated OPC-based composite cured for 90 days. The 1650 cm^−1^ band represents the H-O-H bond deformation vibration and molecular water present in the structure, i.e., the water remaining after the chemical reaction. The SCGs absorbed most of the water during mixing, making the composite cement mix less workable. Even when the quantity of the incorporated coffee waste used was minimal, it affected the setting and hardening of the cement paste. The band observed at 1415 cm^−1^ is associated with the presence of carbonates due to atmospheric CO_2_ interactions with portlandite, and the presence of calcite was also observed on the XRD patterns [117,118,119,120]. The 1112 cm^−1^ band accounts for the Si-O bond asymmetric stretching vibration. The O-Si-O bending vibrations were observed at 450 cm^−1^ [121]; the intensity variation in each cement sample was due to the crystallinity of the materials. The Si-O linkage is associated with Q^n^ units, where n = 0, 1, 2, 3, and 4, and it could be found at 850 cm^−1^, 900 cm^−1^, 950 cm^−1^, 1100 cm^−1^, and 1200 cm^−1^_,_ respectively. The characteristic polymerization of the silicate group in the C-S-H gel, with a strong formation of Q^1^ and Q^2^ tetrahedra and a stretching of Si-O-Si or Al-O-Si, was observed in all the results at 873 cm^−1^ and 960 cm^−1^ [122]. The C-S-H formation was the main factor for the development of compressive strength in the neat OPC and composite cement [123]. The OPC reference sample displayed a higher intensity band than the composite cement, and the bandwidth increased for the composite cement. Broadening of the band was due to the reduction in the silicate polymerization, which is associated with the CaO/SiO_2_ ratio, as observed in the EDS results for the composite cements [117]. The presence of Al_2_O_3_ could also alter these bands. As a result, the increase in the CS implies the development of higher quantities and modification of the C-S-H gel.

Figure 10b shows the ATR infrared spectra obtained for the neat CSA and mixtures, cured for 90 days. As for the OPC composite results discussed above, the H-O-H bond and molecular water bending mode appeared at 1660 cm^−1^ [121,124], with an increase of its intensity for the composite cement prepared with the SCGs and pozzolans as compared to the reference sample. This is due to the free water present in the composite because SCGs occupy most of the water. The bands at 1425–1590 cm^−1^ and 872 cm^−1^ disclosed carbonate presence like the OPC composites [118,119,120]. The stretching vibrations of SO_4_^2−^ (v3) found between 1100 cm^−1^ and 1170 cm^−1^ are associated with the main CSA reaction product: ettringite [113]. The C-A-S-H gel, represented by the presence of the polymerized silicate group, was observed in all the systems at around 1110 cm^−1^, characterized by an asymmetric stretching of the Si-O-Si or Al-O-Si groups [122,124]. The presence of this gel results in the increase of the CS in composite cements [121]. Additionally, stretching of the Si-O-Si bond, depicted by the 522 cm^−1^ bands, is believed to be associated with the C-A-S-H gel. The drop in intensity of that band observed in composite cements SCC3.5, SCFC, and SCVC suggests a decrease in silicate polymerization [118]. The peak width increased for the composite cement prepared with SCGs and pozzolans, indicating that the presence of SCGs affected the formation of the hydration product, which subsequently led to a decrease in the compressive strength (CS) for these blends.

### 3.5. Hydration Temperature Profiles Measurements

Figure 11 illustrates the hydration temperature profiles over time (24 h) for (a) the OPC and (b) CSA composite cements. Xiaotian Zou carried out a similar experimental setup to analyze the hydration temperature of concrete [63]. The hydration process is an exothermic reaction resulting from hydration reactions when cement is mixed with water. The liberation of heat during the cement hydration process is known as the heat of hydration [125]. During this hydration, binding gels are formed and result in the setting and hardening of cement [126,127]. The heat released during the hydration process depends on the type of cement used, the water–cement ratio, cement fineness, curing temperature, additives, or Appendix A used [128]. The heat release is the highest within the first 24 h of setting.

Figure 11a shows the temperature evolution in the OPC and OPC-based composite cements. During the first 4 h, there was an intense chemical activity corresponding to the dissolution and formation of several ionic species. The first few minutes could be associated with period 1, the dissolution period. The next hours were ‘the induction period’, (period 2) where low activity was observed. The main hydration peak was recorded from 4 to 24 h (period 3) [129,130,131]. The initial setting starts at the end of the induction period (from where the temperature rises). The final set is just before the peak temperature. This is the region where the C-S-H and CH reaction products form [64,132]. OC0 and OCF30 showed the maximum temperatures of 43 °C and 42 °C, respectively, at 10.45 and 11.50 h of the hydration process. For the OCV30 sample at 12.00 h, the peak of hydration was at 37 °C. A similar decrease in the hydration temperature was observed in studies conducted by Al. Fadala [41]. Their results indicate that the introduction of pozzolans FA, VA, and SCGs affected the acceleration–deceleration period of the composite cements [128] and that the partial replacement of the OPC with SCGs (OCC3.5) further delayed the hydration process from 10.45 h to 13.20 h, with a measured hydration temperature of 27 °C. The combined effect of delaying the acceleration period and prolonging the peak of the hydration curve means that the reaction is slower. During hydration, a large amount of Ca^2+^ ions are released, and the transport rate of Ca^2+^ ions is higher than that of the SiO_4_^2−^ ion group, promoting crystalline CH formation [67]. The presence of SCGs affected the ionic transport, therefore delaying the formation of the hydration products. It also affected the final setting time in the composite cement samples. The partial replacement of the OPC with SCGs and pozzolans curtailed the hydration temperature to 25 °C within the initial hours, and then it was maintained at a constant range. The substitution of the cement with SCGs dropped the temperature by about 41%. The composite cement prepared using the OPC and SCGs therefore required temperature assistance to set and harden [133].

Figure 11b shows the hydration temperature profile over time for the CSA composite cement. The hydration reaction in the CSA is more exothermic than the one in the OPC. An exothermic peak with a broad shape was observed, indicating higher reactivity reaching up to 64 °C, with the final setting time for the reference sample SC0 at 10:45 h. As discussed earlier, hydration reaction products are formed in the induction period from 4 h to 24 h; here, mainly ettringite and monosulfate precipitated [134]. The hydration performance decreases steadily with the heat flux dropping to room temperature at 20 h. The partial replacement of the CSA with FA led to a temperature of 54 °C and delayed the final set to 11:15 h, while the VA delayed the set by up to 12 h and dropped the temperature to 45 °C. In the case of SCGs (SCC3.5), the hydration process was delayed for 3 more hours, with the final set at 13:45 h. The addition of pozzolans lowered the hydration temperature, as in the case of the OPC-based composite cement. The presence of additional surface areas through the introduction of the FA accelerated nucleation and the formation of hydrates [28]. The presence of SCGs in SCVC and SCFC produced a rise in temperature within the initial 3 h; for SCVC, the temperature was 35 °C and for SCFC, 39 °C. It is shown that the level of substitution of the cement with SCGs influenced the starting time of the acceleration period and the duration of the acceleration and deceleration periods. The results indicate that this replacement level accounted for approximately a 37% decrease in peak heat flux; the latent period was prolonged, and the onset of the acceleration period was delayed. The combined effect of delaying the acceleration period and prolonging the peak of the hydration curve means that the reaction was slower.

It is evident from Figure 11 that the addition of SCGs reduced the hydration temperature and delayed the hydration process in both the OPC and CSA systems [51]. Furthermore, the presence of SCGs reduced the hydration temperature of the composite cement containing FA and VA. However, even though the SCGs decreased the hydration temperature of the composite cement, the drop did not affect the setting and hardening of the cement paste.

## 4. Conclusions

The main findings resulting from this study are outlined below:Composite cement blended with pozzolans attained maximum strength at later ages; this implies the slow reaction of pozzolans. After 210 days of curing, the compressive strength (CS) for the OPC-incorporated composite cement was 42.2 MPa and 45 MPa for OCF30 and OCV30, respectively. For the CSA-incorporated composite cement, the CS was 53.6 MPa for SCV30 and 43 MPa for SCF30, after 210 days of curing. These values suggest that composite cement mixes can be used for structural works.An increase in the percentage of SCGs caused a reduction in workability and CS and delayed setting and hardening. The CS results of the SCG composite cement also show that they can be used for non-structural applications. The addition of SCGs in the composite cement mix reduced the hydration temperature, which is related to the heat adsorption capacity of SCGs; hence, it can be used as a retarder in cement mixes.The characterization techniques, XRD, SEM-EDS, and ATR, revealed the presence of hydration products, C-S-H, portlandite, ettringite, and C-A-S-H, which caused an increase in the CS for the composite cements.The fineness of VA resulted in increased CS values in the composite cement blends. The round spherical shape of FA occupies spaces in the cementitious matrix and leads to a filler effect. When comparing the CS results of the FA and VA for either OPC- and CSA-based compositions, the VA performed better than FA, and this indicates that VA has more pozzolanic activity than FA.After comparing all the data in this work, it was concluded that replacing the OPC with CSA would provide a foundation for the reduction in OPC consumption and, hence, mitigate CO_2_ emissions. Non-structural applications and pavements could benefit from the partial replacement of CSA with SCGs.

## 5. Future Works

In this work, the CSA-VA composite cement showed the best CS results. The replacement of CSA by 70–80% with VA and other pozzolans will lead to economic and environmental benefits. The hybrid cement utilizing CSA and VA will be innovative, giving maximum CS values and less CO_2_ emission. Hence, an investigation focusing on the strength and microstructural studies of hybrid cement made with CSA, calcinated SCGs, VA, FA, and S would attract the attention of the scientific and industrial communities.

## Figures and Tables

**Figure 1 materials-16-05006-f001:**
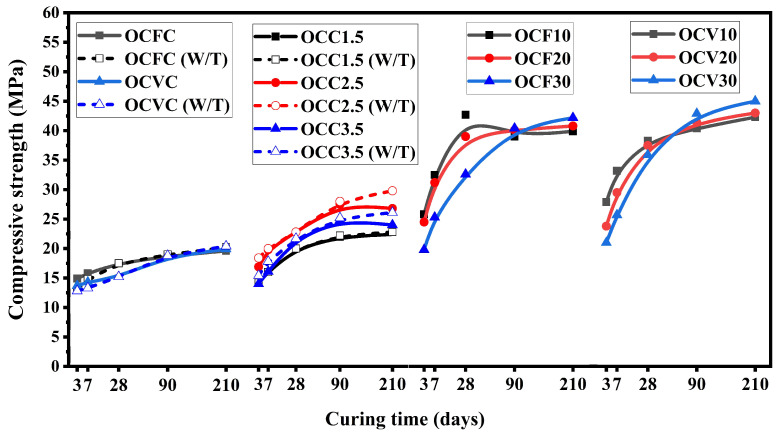
Compressive strength results for the OPC composite cements at different curing ages.

**Figure 2 materials-16-05006-f002:**
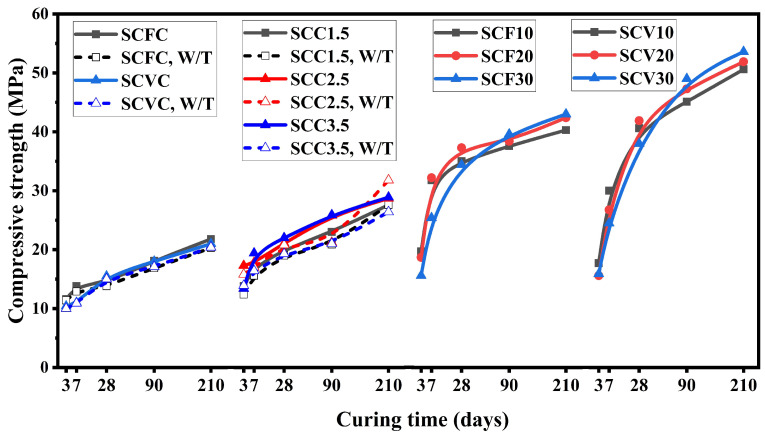
Compressive strength results for the CSA composite cements at different curing ages.

**Figure 3 materials-16-05006-f003:**
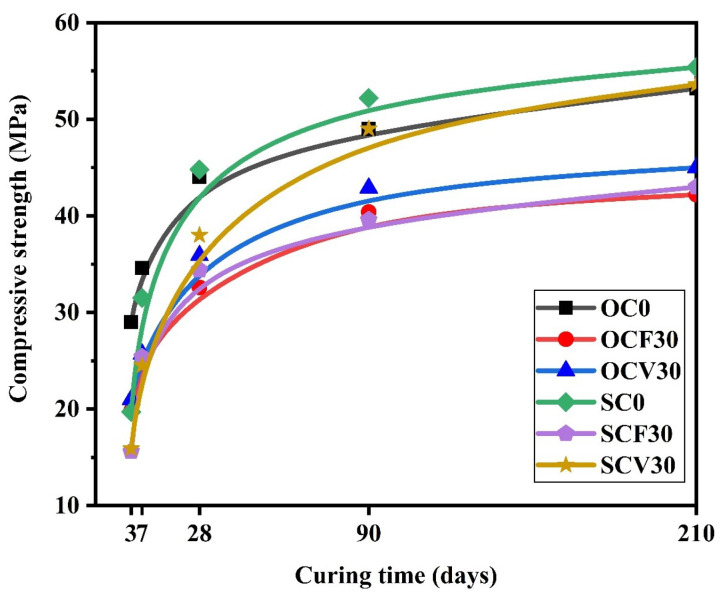
Comparison of compressive strength results for the OPC and CSA composite cement at different curing times.

**Figure 4 materials-16-05006-f004:**
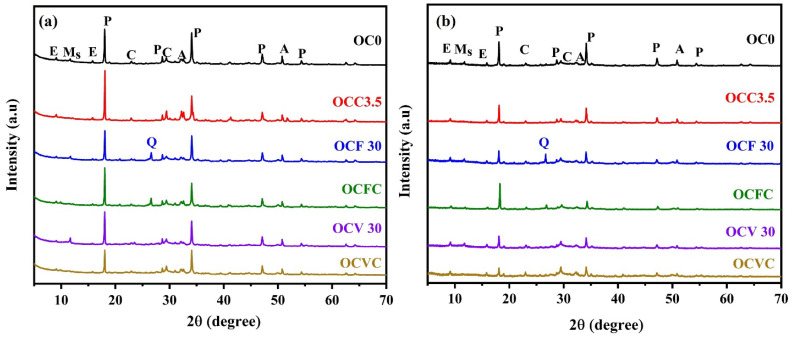
Diffraction patterns of the OPC and its composite systems cured at (**a**) 28 days and (**b**) 90 days of curing. A = alite, C = calcite, E = ettringite, M_s_ = monosulfate, P = portlandite, Q = quartz.

**Figure 5 materials-16-05006-f005:**
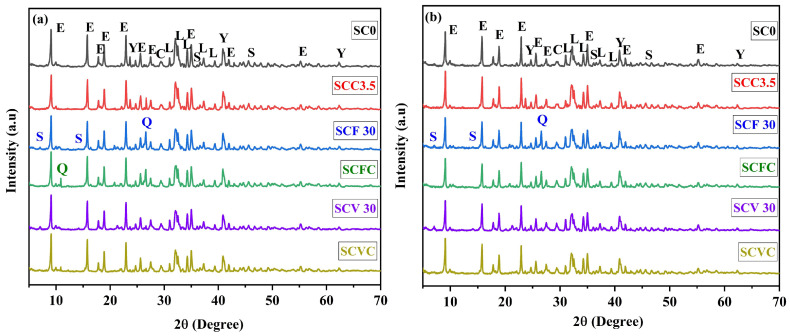
Diffraction patterns of CSA and its composite systems cured at (**a**) 28 days and (**b**) 90 days of curing. C = calcite, E = ettringite, L = larnite, Y = ye’elimite, Q = quartz, S = strätlingite.

**Figure 6 materials-16-05006-f006:**
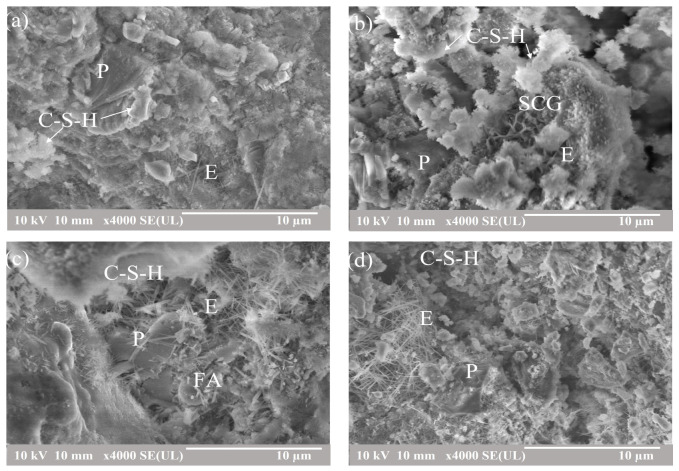
SEM-SE images at 4000× magnification of the reference and composite cement samples (**a**) OPC, (**b**) OCC 3.5, (**c**) OCFC, and (**d**) OCVC. P = portlandite, E = ettringite, C-S-H = calcium silicate hydrate.

**Figure 7 materials-16-05006-f007:**
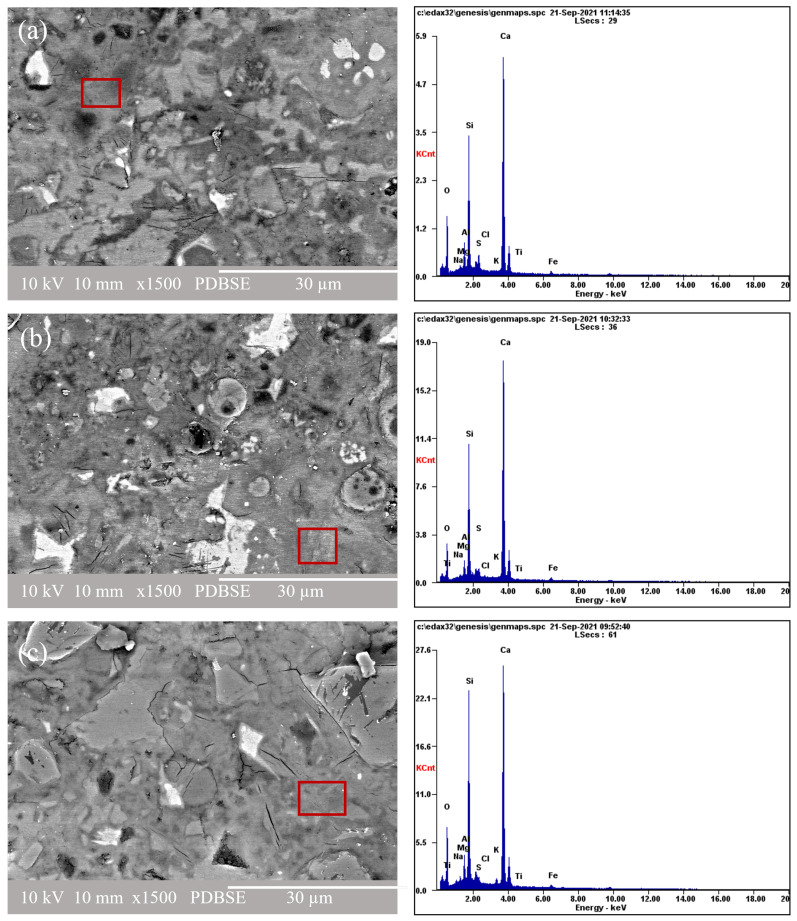
The SEM-BEI images at 1500× magnification and the EDS spectra of the composite cement based on the OPC at 90 days of curing: (**a**) OCC 3.5, (**b**) OCVC, (**c**) OCFC.

**Figure 8 materials-16-05006-f008:**
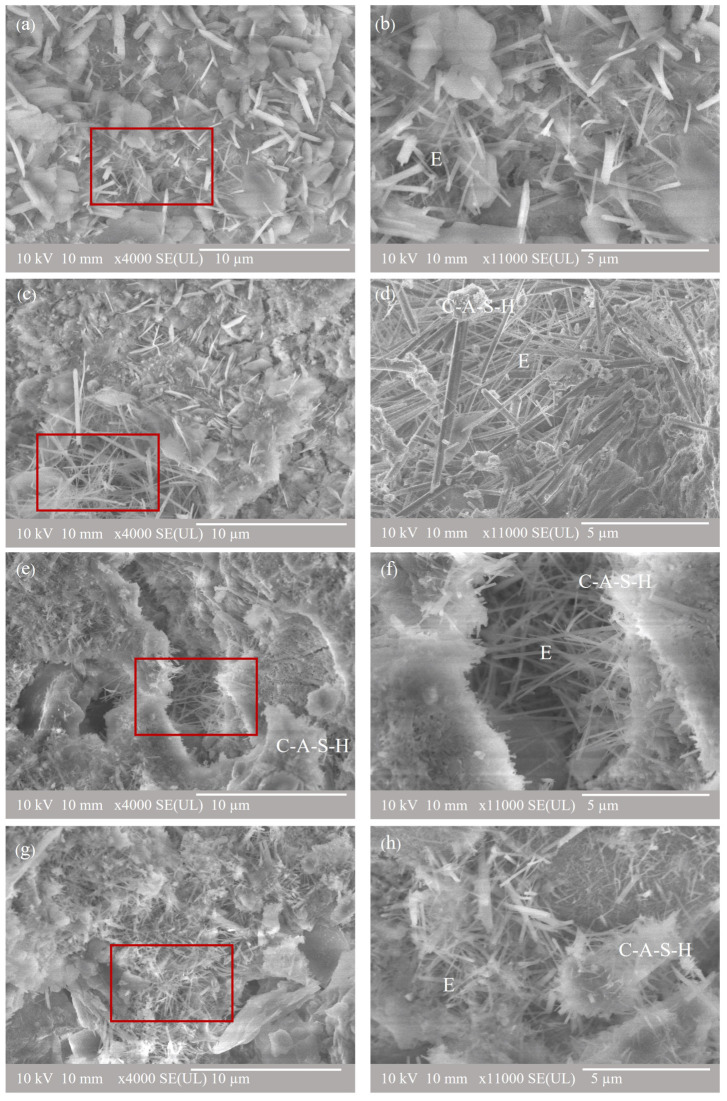
SEM-SE images at 4000× and 11,000× magnification of the reference and composite cement samples: (**a**) CSA, (**b**) CSA, (**c**) SCC 3.5, (**d**) SCC 3.5, (**e**) SCFC, (**f**) SCFC, (**g**) SCVC, (**h**) SCVC. E = ettringite, C-A-S-H = calcium alumino-silicate hydrate.

**Figure 9 materials-16-05006-f009:**
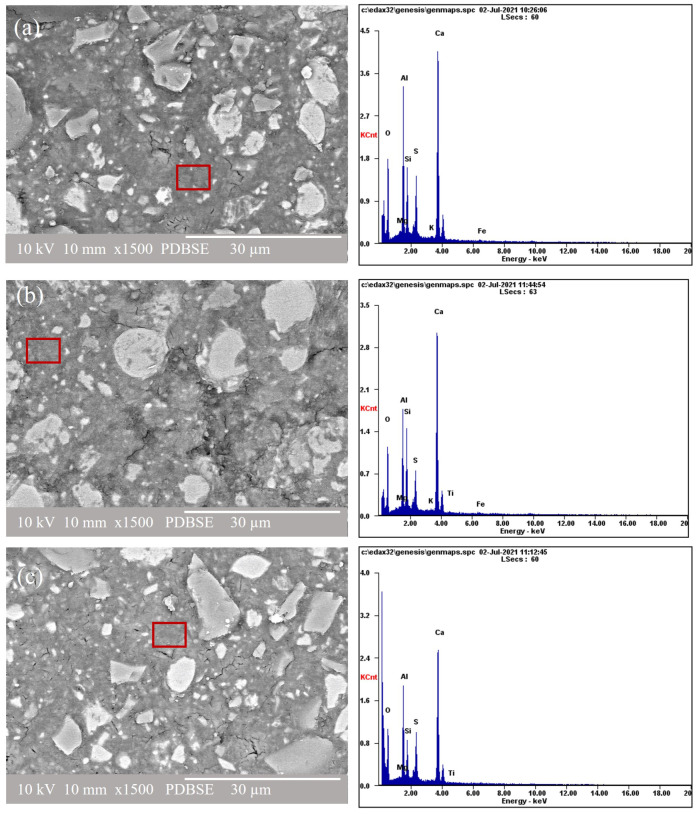
The SEM-BEI images at 1500× magnification and the EDS spectra of the composite CSA cement samples cured at 90 days: (**a**) SCC 3.5, (**b**) SCFC, (**c**) SCVC.

**Figure 10 materials-16-05006-f010:**
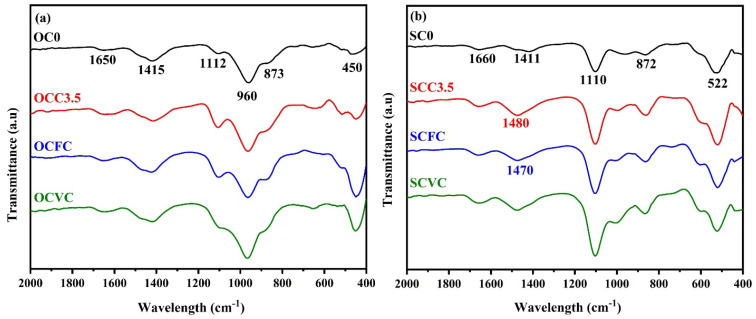
ATR infrared spectra of (**a**) OPC and (**b**) CSA and the corresponding composite cement, cured for 90 days.

**Figure 11 materials-16-05006-f011:**
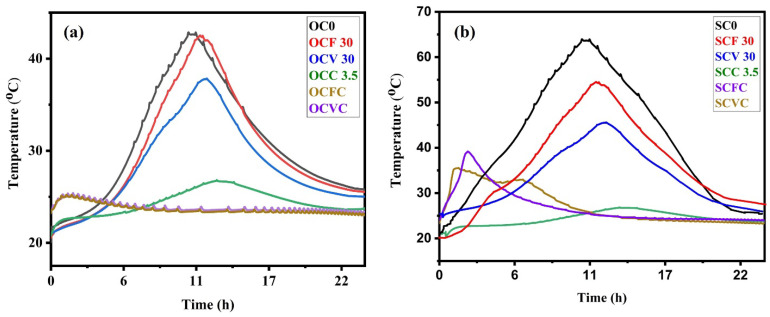
(**a**) Hydration temperature of the OPC composite cement pastes, (**b**) hydration temperature of the CSA composite cement pastes.

**Table 1 materials-16-05006-t001:** Chemical composition (XRF) and density of raw materials.

Composition	OPC	CSA	FA	VA	SCG
CaO	63.17	55.1	5.85	2.31	32.89
SiO_2_	17.68	12.51	76.47	73.29	3.08
Al_2_O_3_	3.94	15.05	11.51	14.83	2.66
Fe_2_O_3_	3.34	0.95	2.38	0.92	9.54
SO_3_	3.21	13.88	1.44	0.68	13.71
MgO	1.13	0.88	0.37	0.32	6.78
K_2_O	0.84	0.71	0.87	4.83	22.57
Na_2_O	0.25	-	0.15	0.09	-
TiO_2_	0.22	0.98	0.56	0.1	-
MnO	0.12	-	0.012	0.15	0.52
P_2_O_5_	0.11	-	-	-	4.93
SrO	-	0.12	0.024	-	0.34
CuO	-	0.04	-	-	0.97
Cr_2_O_3_	-	0.03	-	-	1.52
ZnO	-	0.02	0.0056	-	0.27
ZrO_2_	-	0.02	-	-	-
V_2_O_5_	-	0.02	0.013	-	-
Density (g/cm^3^)	3.12	2.89	2.1	2.37	1.39

**Table 2 materials-16-05006-t002:** Mixture design of composite cement mixtures prepared with the OPC.

Composition	Cement Replacement Percentage
OC0	0
OCV10	10% VA
OCV20	20% VA
OCV30	30% VA
OCF10	10% FA
OCF20	20% FA
OCF30	30% FA
OCC1.5	1.5% SCG
OCC2.5	2.5% SCG
OCC3.5	3.5% SCG
OCFC	30% FA, 3.5% SCG
OCVC	30% VA, 3.5% SCG
OCC1.5 (W/T)	1.5% SCGs, made at the temperature of 80 °C.
OCC2.5 (W/T)	2.5% SCGs, made at the temperature of 80 °C.
OCC3.5 (W/T)	3.5% SCGs, made at the temperature of 80 °C.
OCFC (W/T)	30% FA, 3.5% SCGs, made at the temperature of 80 °C.
OCVC (W/T)	30% VA, 3.5% SCGs, made at the temperature of 80 °C.

**Table 3 materials-16-05006-t003:** Mixture design of composite cement prepared with the CSA.

Composition	Cement Replacement Percentage
SC0	0
SCV10	10% VA
SCV20	20% VA
SCV30	30% VA
SCF10	10% FA
SCF20	20% FA
SCF30	30% FA
SCC1.5	1.5% SCGs
SCC2.5	2.5% SCGs
SCC3.5	3.5% SCGs
SCFC	30% FA, 3.5% SCGs
SCVC	30% VA, 3.5% SCGs
SCC1.5 (W/T)	1.5% SCGs, made at the temperature of 80 °C.
SCC2.5 (W/T)	2.5% SCGs, made at the temperature of 80 °C.
SCC3.5 (W/T)	3.5% SCGs, made at the temperature of 80 °C.
SCFC (W/T)	30% FA, 3.5% SCGs, made at the temperature of 80 °C.
SCVC (W/T)	30% VA, 3.5% SCGs, made at the temperature of 80 °C.

**Table 4 materials-16-05006-t004:** Oxide composition, in %, for composite cements calculated from the EDS analysis.

Composites	Oxides (%)				
	CaO	SiO_2_	Al_2_O_3_	Na_2_O	SO_3_
OCC3.5	54.83	37.2	6.95	1.02	3.86
OCFC	47.42	41.09	10.76	0.73	4.2
OCVC	47.97	42.48	8.27	1.28	3.66
SCC3.5	46	14.27	39.53	0.21	24.14
SCFC	46.8	15.2	38.67	0.2	17.96
SCVC	46.71	14.56	38.53	0.2	29.33

## Data Availability

Not applicable.

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
