# Peer review of "Strength and Microstructure Assessment of Partially Replaced Ordinary Portland Cement and Calcium Sulfoaluminate Cement with Pozzolans and Spent Coffee Grounds"

_materials, 2023, doi:10.3390/ma16145006_

Round 1

Reviewer 1 Report

materials-2471067-Review comments:

Reducing CO2 emissions from cement has been a widespread concern. In this manuscript, the study compared the impact of adding spent coffee grounds (SCG), fly ash (FA), and volcanic ash (VA) to two types of cements: OPC and calcium sul- foaluminate cement (CSA). The research object and method have certain innovation. However, there are some problems in the manuscript.

1. In this manuscript, various research methods such as XRD , SEM and ATR are used, but these research methods are simply listed and what is the connection between the research methods is not explained?

2.The title of the manuscript is the evaluation of cement strength by two additives, but there are three in the abstract.

3. Composite cement may be affected by other factors such as temperature and time. How to consider these factors in the experiment?

4.In this manuscript, the data are all from experiments. How to consider the transformation from experiments to engineering?

5. What are the characteristics of OPC and CSA cement in this manuscript?

6. The conclusion section need to be further improved and condensed to highlight the innovation of the research work.

7. There are many format errors and layout problems in the manuscript, such as the Table4Fig10 and so on.

8. Text and pictures should be close to each other for easy understanding, such as Figure 6 and Figure 7.

9. In this manuscript, there are too few references in the last three years.

In summary, please revised the manuscript according to the above problems.

barely satisfactory

Author Response

Title: Strength and microstructure assessment of partially replaced ordinary portland cement and calcium sulfoaluminate cement with pozzolans and spent coffee grounds

General response: We highly appreciate the reviewers for the constructive comments and suggestions. According to the referee’s comments, the manuscript has been revised thoroughly. We believe that the revised manuscript has been significantly improved and now meets the criteria for possible publication in the Journal of Materials. Detailed authors’ responses (AR) to the reviewers’ comments are listed below one by one. English language was improved via proofreading and changes made to the manuscript are also detailed via track changes in the attached document.

Reviewer 1

Reducing CO2 emissions from cement has been a widespread concern. In this manuscript, the study compared the impact of adding spent coffee grounds (SCG), fly ash (FA), and volcanic ash (VA) to two types of cements: OPC and calcium sulfoaluminate cement (CSA). The research object and method have certain innovation. However, there are some problems in the manuscript.

Q1. In this manuscript, various research methods such as XRD , SEM and ATR are used, but these research methods are simply listed and what is the connection between the research methods is not explained?

AR: The results obtained from the characterization techniques used in the investigation are correlated. The compressive strength results have a correlation with phases obtained in the XRD results. Here is one example.  Blends OCF30, and OCV30, showed a decrease in the portlandite, suggesting a pozzolanic reaction at later ages, which is in line with the CS results (Lines 345-346). The characteristic phases obtained in XRD pattern were also found in the SEM images, here are some examples. C-S-H, which due to its poorly crystalline structure, could not be identified in XRD results, was visible on the SEM images (Line 400-401). The main hydration product of CSA was ettringite, as observed in figure 8 (SEM image of CSA and its composite cement samples) and XRD results and in agreement with the literature (Line 489-490). The C-A-S-H gel, represented by the presence of polymerized silicate group, was observed in all the systems at around 1110 cm-1, characterized by an asymmetric stretching of Si-O-Si or Al-O-Si [116,118]. The presence of this gel results in the increase of CS for the composite cement (Line 566-569).

Q2. The title of the manuscript is the evaluation of cement strength by two additives, but there are three in the abstract.

AR: We thank reviewer for the suggestion, and we modified the title as “Strength and microstructure assessment of partially replaced ordinary Portland cement and calcium sulfoaluminate cement with pozzolans and spent coffee grounds” (Here the pozzolans are including fly and volcanic ashes).

Q3. Composite cement may be affected by other factors such as temperature and time. How to consider these factors in the experiment?

AR: We appreciate the valuable comment. The hydration temperatures of samples were calculated, which was explained in section 3.5 (Hydration temperature profiles measurements). From the experiment it was concluded that the addition of pozzolans FA, VA, SCG affected the peak hydration temperature and delayed the setting time, as shown in Fig. 11.

Q4. In this manuscript, the data are all from experiments. How to consider the transformation from experiments to engineering?

AR: Paper established that results could be used in non-structural work in general, with some results for structural works. Comparing the CS results of composite cement with OPC and CSA, the composite cement comprising CSA and VA outperformed all other composite cement made with OPC. After comparing all data, it was determined that replacing OPC with CSA would provide a foundation for reducing OPC consumption and, hence, reducing CO2 emissions. Non-structural applications and pavements could benefit from the partial replacement of CSA with SCG. CSA-VA composite cement showed the best CS results in this investigation, this could be used in structural works. Please, see modified conclusions for this info.

Q5. What are the characteristics of OPC and CSA cement in this manuscript?

AR: The chemical composition of OPC and CSA were obtained using XRF characterization which is listed in Table 1. Main differences are stated in the chemical composition. The OPC used in the investigation has 63.17% of CaO, 17.68% SiO2, 3.94% of Al2O3, 3.34% Fe2O3, 3.21% of SO3 and smaller % of other oxides. While CSA has 55.1%CaO, SiO2 12.5%, Al2O3 15.05% SO3 13.88%. Comparing OPC with CSA, CSA have high amount of SO3.The density of OPC was 3,12 and for CSA it was 2.89.

Q6: The conclusion section need to be further improved and condensed to highlight the innovation of the research work.

AR: Conclusion has been improved and modified as follows:

The main findings resulting from this study are outlined below:

  • Composite cement blended with pozzolans attained maximum strength at later ages; this implies the slow reaction of pozzolans. After 210 days of curing, the CS for OPC-incorporated composite cement CS was 42.2 MPa and 45 MPa for OCF30 OCV30, respectively. For CSA-incorporated composite cement, CS was 53.6 MPa for SCV30 and 43 MPa for SCF30 after 210 days of curing. These values suggest that the composite cement mix can be used for structural works.
  • Increase in percentage of SCG caused a reduction in workability and CS, delayed setting and hardening. The CS results of SCG composite cement also show that they can be used for non -structural applications. Addition of SCG in the composite cement mix reduced the hydration temperature, this is related to heat adsorption capacity of SCG, hence it can be used as a retarder in the cement mixes.
  • Characterization techniques XRD, SEM-EDS, ATR, reveal the presence of hydration products C-S-H, portlandite, ettringite, C-A-S-H, which were the reason for increase in CS of the composite cement. Cementitious gels C-S-H and C-A-S-H for OPC and CSA composite cement have been confirmed based on the relationships of the oxides that were present in the matrix of all the systems synthesized in this work.
  • The fineness of VA resulted in increased CS for the composite cement mix. The round spherical shape of FA occupies the spaces in the cementitious matrix leads to a filler effect. Comparing the CS results of FA and VA with each of the cement OPC and CSA, VA performed better than FA, this indicates that VA has more pozzolanic activity than FA.
  • After comparing all data in this work, it was concluded that replacing OPC with CSA would provide a foundation for reducing OPC consumption and, hence, reducing CO2 Non-structural applications and pavements could benefit from the partial replacement of CSA with SCG.

Q7. There are many format errors and layout problems in the manuscript, such as the Table4、Fig10 and so on.

AR: Manuscript has been revised in detail and corrected accordingly.

Q8. Text and pictures should be close to each other for easy understanding, such as Figure 6 and Figure 7.

AR: Manuscript has been revised and corrected.

Q9. In this manuscript, there are too few references in the last three years.

AR: Relevant recent references were added in the manuscript [1][2][3][4]

Reviewer 2 Report

Typo are seen everywhere. Professional proofread is required. 

Author Response

Comments:

Q1: “Strength assessment of partially replaced ordinary Portland cement and calcium sulfoaluminate cement with pozzolans and spent coffee grounds” should be revised for better representations as “Strength and microstructure assessment of partially replaced ordinary Portland cement and calcium sulfoaluminate cement with pozzolans and spent coffee grounds.”

AR: The title has been modified as per the recommendation of the reviewer.

Q2: Line 29-30: This can be unclear statement (C-A-S-H phase and ettringite should be precisely defined since ettringite is C-A-S-H)

AR: We agree with reviewer’s point and text has been modified

Q3: Keywords: the terms ordinary Portland cement, pozzolans, spent coffee grounds should be addressed

AR: Keywords were modified accordingly.

 Q4: Introduction: line 34-35 the statement is unclear and should be revised as: Cement is the most widely used man made building material in the world, and as the population grows, demand also surges significantly.

AR: This sentence has been revised for clarity.

Q5: Line 42: Supplementary cementing material should be revised to SCM here

AR: Line 42 has been corrected accordingly.

Q6: Line 43: blast furnace slag (S), fly ash (FA), kaolin… this sentence was repeated.

AR: The repeated sentence is deleted.

Q7:Line 43: Rice husk ash (RHA) should be first introduced here before line 124. Relevant references should be added. Suggested references include:

L.Prasittisopin, D. Trejo(2013) Characterization of chemical treatment method for rice husk ash cementing materials, ACI SP294,1-14

A-Mansour, A., Chow, C.L.,Feo, L., Penna, R., &Lau, D. (2019). Green concrete: By- products utilization and advanced approaches. Sustainanbility, 11(19), 5145

AR:  Line 43 is modified accordingly  and relevant references have now been added to the manuscript text [5,6].

Q8: Table 1: Revise “SAC” to “CSA”

AR: The reviewer is correct; as such, relevant correction has been made.

Q9: Table 2 and 3  Term “W/T” should address the 80℃ in the explanation

AR: The table 2 and 3 have been modified by mentioning the temperature 80℃

Q10: Section 2.1 (Line 159-174) VA typically has high water absorption. The authors should identify the flow of each system since the w/c was fixed.

AR:  The manuscript has been revised, and relevant corrections have been made.

For optimizing the w/c, OPC and CSA incorporated composite cement were prepared separately with each FA, VA, and SCG at different w/c (0.4, 0.45, 0.5). It was found that FA and VA require comparatively less w/c than SCG. Also, increasing the w/c ratio for composite cement samples with FA and VA decreases the compressive strength (CS). While using w/c 0.4, there was a shrinkage in the hardened state for FA and VA incorporated composite cement; at 0.4 for SCG, the samples required extra water to make the composite cement workable. Raising the w/c to 0.5 decreases CS results for all the composite cement with FA, VA, and SCG and show a delay in setting and hardening for composite cement with SCG. At w/c 0.45, all the composite cement samples were workable, and VA showed better CS results than FA and SCG.

Q11: Line 131-132: “As alternatives to OPC, calcium sulfoaluminate cement and FA-based geopolymer mortar were studied in the past to evaluate…”: This sentence should be revised to “As alternatives to OPC, CSA mortar and CSA FA-based geopolymer mortar were studied in the past to evaluate..” The reference about the CSA mortar  should be introduced. Suggested ones include:

Li, Lin, Ru Wang, and Qinyuan Lu. “Influence of polymer later on the setting time, mechanical properties and durability of calcium sulfoaluminate cement mortar.” Construction and Building Materials 169 (2018): 911-922

Sereewatthanawut, Issara, et al. “Chloride- induced corrosion of a galvanized steel-embedded calcium sulfoaluminate stucco system.” Journal of Building Engineering 44 (2021): 103376

AR: Relevant references have now been added to the manuscript text [7,8]

Q12: Conclusion: Line 633-634: compressive strength (CS) has been introduced

AR: Relevant corrections have been made as per the reviewer’s comment.

Q13: Line 637: Quantitative value should be provided

AR: Quantitative values were added as per the reviewer’s comment

Composite cement blended with pozzolans attained maximum strength at later ages; this implies the slow reaction of pozzolans. After 210 days of curing, the CS for OPC-incorporated composite cement CS was 42.2 MPa and 45 MPa for OCF30 OCV30, respectively. For CSA-incorporated composite cement, CS was 53.6 MPa for SCV30 and 43 MPa for SCF30 after 210 days of curing. These values concludes that the composite cement mix can be used for structural works.

Q14: Line 667: Future recommendation works should be  discussed.

AR: Future work added as per reviewer’s suggestion

In this work, CSA-VA composite cement showed the best CS results. The re-placement of CSA by 70-80% with VA and other pozzolans will lead to economic and environmental benefits. The hybrid cement utilizing CSA and VA will be innovative, giving maximum CS and less CO2 emission. Hence the investigation on strength and microstructure study of hybrid cement with CSA, calcinated SCG, VA, FA, and S would attract the attention of the scientific and industrial community.

Reviewer 3 Report

This manuscript is a well-written manuscript with experimental results.

It would be great if the novelty of this research is emphasized in the last part of the introduction.

Also, a photo of the spent coffee grounds and a photo of pozzolans are recommended to be attached to the manuscript.

Line 43: “F” should not be uppercase from the word “Fly ash”.

Lines 148-152: Please, check uppercase and lowercase in the sentences.

Lines 152-155: Are there any standard test methods (ex., ASTM, BS/EN, etc.) the authors employed?

Line 203: It is recommended to be “Table 2” instead of “table 2”.

Table 3: It should be “30% VA” instead of “30%VA” for SCVC and SCVE(W/T).

Figures 1, 2, and 3 must be re-designed. It looks like it is manufactured by Photoshop.

Grammer should be checked.

Author Response

Q1. It would be great if the novelty of this research is emphasized in the last part of the introduction.

AR: Very little to no research has been done using VA and CSA composite cement so far; this research report on using VA with CSA blended mix for the first time could be used for structural applications. This is the first report of SCG in a composite cement mix for non-structural applications. The document now established this information.

Q2. Also, a photo of the spent coffee grounds and a photo of pozzolans are recommended to be attached to the manuscript.

AR: The images of pozzolans FA,VA and SCG are attached in the supplementary file S1.

Q3. Line 43: “F” should not be uppercase from the word “Fly ash”.

AR: The text has been corrected in the manuscript.

Q4. Lines 148-152: Please, check uppercase and lowercase in the sentences.

AR: The text has been corrected in the manuscript.

Q5. Lines 152-155: Are there any standard test methods (ex., ASTM, BS/EN, etc.) the authors employed?

AR: Yes, we have used ASTM 0402C311, ASTM0401C595 standards were employed for the preparation of raw materials.

Q6. Line 203: It is recommended to be “Table 2” instead of “table 2”.

AR: The text has been corrected in the manuscript.

Q7. Table 3: It should be “30% VA” instead of “30%VA” for SCVC and SCVE(W/T).

AR: The text has been corrected in the manuscript.

 Q8. Figures 1, 2, and 3 must be re-designed. It looks like it is manufactured by Photoshop.

AR:  None of the images were prepared using photoshop. Figures 1, 2, and 3 were drawn using Origin, with the b-spline function. Here we are providing origin embedded images of Figure 1, 2, and 3 for your reference.

Figure 1: The compressive strength results of the OPC composite cement at different curing ages

Figure 2. The compressive strength results of the CSA composite cement at different curing ages

Figure 3. Comparison of compressive strength results for the OPC and CSA composite cement at different curing times.

REFERENCES FOR ALL THE REVIEWER’S COMMENTS:

  1. Tao, Y.; Rahul, A.V.; Mohan, M.K.; De Schutter, G.; Van Tittelboom, K. Recent Progress and Technical Challenges in Using Calcium Sulfoaluminate (CSA) Cement. Cem. Concr. Compos. 2023, 137, 104908, doi:10.1016/j.cemconcomp.2022.104908.
  2. Kim, T.; Seo, K.-Y.; Kang, C.; Lee, T.-K. Development of Eco-Friendly Cement Using a Calcium Sulfoaluminate Expansive Agent Blended with Slag and Silica Fume. Appl. Sci. 2021, 11, 394, doi:10.3390/app11010394.
  3. Liao, Y.; Gui, Y.; Wang, K.; Al Qunaynah, S.; Bawa, S.M.; Tang, S. Activation Energy of Calcium Sulfoaluminate Cement-Based Materials. Mater. Struct. 2021, 54, 162, doi:10.1617/s11527-021-01753-3.
  4. Aguilera, C.; Viteri, M.; Seqqat, R.; Ayala Navarrette, L.; Toulkeridis, T.; Ruano, A.; Torres Arias, M. Biological Impact of Exposure to Extremely Fine-Grained Volcanic Ash. J. Nanotechnol. 2018, 2018, 1–12, doi:10.1155/2018/7543859.
  5. Lapyote, Prasittisopin; Trejo, D. Characterization of Chemical Treatment Method for Rice Husk Ash Cementing Materials. In Proceedings of the SP-294: Advances in Green Binder Systems; American Concrete Institute, 2013.
  6. Al-Mansour, A.; Chow, C.L.; Feo, L.; Penna, R.; Lau, D. Green Concrete: By-Products Utilization and Advanced Approaches. Sustain. 2019, 11, doi:10.3390/su11195145.
  7. Li, L.; Wang, R.; Lu, Q. Influence of Polymer Latex on the Setting Time, Mechanical Properties and Durability of Calcium Sulfoaluminate Cement Mortar. Constr. Build. Mater. 2018, 169, 911–922, doi:10.1016/j.conbuildmat.2018.03.005.
  8. Sereewatthanawut, I.; Pansuk, W.; Pheinsusom, P.; Prasittisopin, L. Chloride-Induced Corrosion of a Galvanized Steel-Embedded Calcium Sulfoaluminate Stucco System. J. Build. Eng. 2021, 44, 103376, doi:10.1016/j.jobe.2021.103376.
